# Retarding Effect of Hemp Hurd Lixiviates on the Hydration of Hydraulic and CSA Cements

**DOI:** 10.3390/ma16165561

**Published:** 2023-08-10

**Authors:** Donato Tale Ponga, Amirmohammad Sabziparvar, Patrice Cousin, Lina Boulos, Mathieu Robert, M. Reza Foruzanmehr

**Affiliations:** 1Department of Civil & Building Engineering, University of Sherbrooke, Sherbrooke, QC J1K 2R1, Canada; donato.tale.ponga@usherbrooke.ca (D.T.P.); patrice.cousin@usherbrooke.ca (P.C.); lina.boulos@usherbrooke.ca (L.B.); mathieu.robert2@usherbrooke.ca (M.R.); 2Department of Civil Engineering, University of Ottawa, Ottawa, ON K1N 6N5, Canada; asabz048@uottawa.ca

**Keywords:** hemp hurd, Portland cement, CSA cement, hydration process, cementitious composites, bio-aggregate lixiviate

## Abstract

Wood wool panels are widely used in the construction industry as sustainable cementitious composites, but there is a growing need to replace traditional Portland cement with a binder that has a lower embodied carbon footprint. In addition, the sustainability of these panels may face serious impediments if the required amount of wood for their production needs a harvest rate higher than the rate at which the tree sources reach maturity. One solution is to use the wooden part of fast-growing plants such as hemp. However, the compounds extracted from the mixture of plants and water are the main cause of the delay observed during the hydration process of hydraulic binders in these cementitious composites. The objective of this study is to evaluate the effect of bio-aggregate lixiviates (hemp hurd) on the hydration kinetics of calcium sulfoaluminate (CSA) cement as a low-embodied-carbon alternative to ordinary Portland cement (OPC). The isothermal calorimeter showed that the hemp hurd lixiviate caused a greater delay in GU’s hydration process than CSA’s. At a 5% concentration, the main hydration peak for GU cement emerged after 91 h, whereas for CSA cement, it appeared much earlier, at 2.5 h. XRD and TGA analysis showed that after 12 h of hydration, hydration products such as calcium silicate hydrates (C-S-H) and portlandite (CH) were not able to form on GU cement, indicating low hydration of silicate products. Moreover, at 5% concentration, the carbonation of ettringite was observed in CSA cement. The compressive strength values obtained from the mixes containing hemp hurd lixiviate consistently showed lower values compared to the reference samples prepared with distilled water. Furthermore, the CSA samples demonstrated superior compressive strength when compared to the GU samples. After 28 days of hydration, the compressive strength values for CSA cement were 36.7%, 63.5% and 71% higher than GU cement at a concentration of 0.5%, 2% and 5% hemp hurd lixiviate, respectively.

## 1. Introduction

During their tenth annual meeting, the United Nations Environment Program committee raised concerns about the intensifying CO_2_ emissions being produced by the construction and building industries. According to statistical studies, this sector emitted ten billion metric tons of CO_2_ in 2019 [1]. Therefore, construction companies must commit to further reducing their CO_2_ emissions. The use of non-renewable materials with high-carbon footprints in the building industry is one of the main contributors to the accumulation of CO_2_ in the atmosphere [2,3]. Therefore, the development of bio-based building materials such as bio-based cementitious composites could be a potential solution. Additionally, cementitious composites modified with supplementary cementitious materials such as fly ash or silica fume have the potential to reduce CO_2_ emissions in the atmosphere. He et al. [4] conducted a study on the influence of sodium chloride and gypsum on the compressive strength and sulfate resistance of slag-based geopolymer concrete. They found that by replacing 32.5% of slag Portland cement with sodium chloride, gypsum, and slag at proportions of 4:7, 5:13, and 5:75 in plain concrete, the cost and carbon emissions of geopolymer concrete were reduced by 25% and 48%, respectively.

In another study, Golewski [5] evaluated the effects of coal fly ash and nanosilica on the strength parameters and microstructural properties of eco-friendly concrete. The research demonstrated that using tailored blended cements composed of nanosilica and coal fly ash, with content up to 30% replacement level, significantly improved the parameters of the concrete composite. Moreover, this approach helped reduce the carbon footprint of cement-based materials, representing a positive step towards the production of eco-friendly concretes.

Sustainable construction sectors have shown increased interest in using bio-based building materials as alternatives to conventional materials. These bio-based materials offer effective hygrothermal properties, making them an attractive option for environmentally friendly construction practices. Cementitious composites incorporating bio-based aggregates have an additional interesting characteristic that makes them an excellent candidate for net-zero building applications: their “thermal mass”. This property is due to the high thermal capacity of the cementitious matrix [6]. These composites can effectively buffer daily temperature variations by absorbing heat and releasing it back into the indoor environment as needed, most often at night. One of the popular bio-based cementitious building materials that has been in use for decades is the wood wool panel. This product, also known as a wood wool cement board (WWCB), is a composite building material made of wood fibres, Portland cement, and water [7,8,9]. While wood, the main component of WWCB, is technically renewable, there are concerns about the rate at which it is harvested for WWCB production. If the rate at which trees are being cut down exceeds the rate at which they are growing, wood cannot be considered a truly renewable material [10].

The use of plant aggregates can be a crucial remedy for the sustainable development of WWCB. Examples of plant aggregates that have been used include straw bales [11] and hemp hurds [12]. Despite their abundance and effective properties, the limited compatibility between plant aggregates and cementitious binders has posed challenges in the manufacturing of WWCB, and ultimately the performance of these composites, using these alternatives [6,8,9,13]. Furthermore, studies have shown that composites made with cement and plant aggregates often have low mechanical strength. This can be attributed to inadequate binder hydration due to the incompatibility between the binders and the vegetal aggregate [14,15,16]. The presence of saccharides in lignocellulosic materials can cause this effect [17]. For example, natural fibre such as hemp consists of diverse saccharides that exhibit varying solubility levels in water. Some of these saccharides can dissolve, resulting in the release of lixiviate, which could negatively influence the properties of the composite [7,18,19].

During the mixing and processing of natural, fibre-reinforced cementitious composites, extractive molecules migrate into the mixwater. This can kinetically inhibit the hydration of binders [20,21,22]. Two mechanisms have been identified to explain the interactions between plant molecules and mineral phases: the adsorption of monosaccharides onto anhydrous cement grains and the elimination of calcium ions facilitated by alkaline-induced organic acids.

Monosaccharides, also known as reducing sugars, can release ions in an alkaline environment. These ions can adsorb on the positively charged surface of cement grains through the formation of van der Waals bonds [23] The reversible physisorption process limits the ability of non-hydrated particles to access water, which in turn delays the setting time of the Portland cement paste. Furthermore, the elimination of calcium ions facilitated by alkaline-induced organic acids is another mechanism that influences the cement hydration process. Organic acids such as acetic, formic, and lactic acids are produced when saccharides are decomposed in an alkaline environment [22,23,24,25]. These acids can scavenge calcium ions, making them unavailable for hydration reactions [26]. In other words, non-reducing sugars are capable of forming strong complexes with calcium ions and preventing the precipitation of the calcium silicate hydrates (C-S-H) phase at the nucleation sites [27]. Moreover, it has also been highlighted by Sedan et al. [12] that pectin can be problematic in cementitious matrices. Pectin has an affinity with calcium ions, forming a non-reactive gel. If pectin reacts with calcium ions, then the calcium ions are not available for hydration, leading to a retarding effect in the composite’s strength development.

For example, Simatupang [28] reported that gluco-saccharic acid, gluco-meta-saccharic acid, lactic acid, and mannose inhibited the formation of C-S-H gel in OPC. Kochova et al. [18,29] conducted a study in which they identified organic acids, such as uranic acids, that exhibited inhibitory effects on OPC hydration and negatively affected the formation of cement hydration products. In addition, citric, tartaric, and gluconic acids, and their salts, are strong retarders in high alumina cement or CSA cement [27,30,31]. The citric acid and gluconate were shown to effectively retard the early hydration reactions, such as the ettringite formation in CSA cement [31,32]. Cody et al. [33] demonstrated the effectiveness of hydroxylated carboxylic acids in inhibiting the nucleation and growth of ettringite. Furthermore, several studies investigated the effect of sucrose and glucose on Portland cement hydration by using calorimeters. These studies revealed that sucrose had a strong retarding effect on the hydration of OPC for up to several months [14,34].

Despite the growing awareness of the interactions between plants and mineral binders, instances of material disintegration in bio-based cementitious composites are still frequently reported after their use as building materials [19,35]. The performance of these types of cement is influenced by various factors, such as clinkerization, hydration kinetics, microstructure development, and durability. However, many of these factors remain poorly understood and require further investigation to search for new types of cement with reduced calcium carbonate production and lower clinkering temperatures for bio-based insulation materials. CSA cement can be an eco-friendlier alternative to OPC since it emits less carbon dioxide in the manufacturing process when compared to OPC cement. Additionally, the friable nature of CSA cement allows for reduced energy consumption during clinker grinding [36,37].

The main focus of this study is to examine the impact of lixiviate obtained from hemp hurd on the hydration process of CSA cement and OPC. The primary objective is to assess the feasibility of using CSA cement as a substitute for OPC in the production of cementitious composites that incorporate hemp hurd. The primary goal of this study is to investigate the adverse effects of hemp hurd leachates on the hydration process of CSA, which has the potential to serve as a more environmentally friendly substitute for OPC in WWCB applications. The research aims to quantify the extent to which these negative effects can delay the setting-time and reduce the mechanical properties of cementitious wooden panels with respect to the concentration of hemp hurd leachates. Furthermore, this study intends to provide valuable insights to subsequent studies to optimize the pre-treatment methods necessary to ensure the compatibility of hemp hurd with the binders used in the production of WWCB.

## 2. Materials and Methods

### 2.1. Materials

#### 2.1.1. Types of Hemp

In this study, hemp hurds were used to produce lixiviate as shown in Figure 1. Hemp hurds are also known as shiv (about 0.5 mm in length). Hurds are the wooden core of stalks. They were provided by the Emerson Hemp Distribution Company (St. Louis, MO, USA) in Manitoba, Canada.

#### 2.1.2. Types of Cement

Two commercially available types of cement, including General Use (GU, CEM I 42.5R, with a density of 3.15 and specific surface area of 393 m^2^/kg) cement and CSA cement were investigated in the current study. These types of cement were provided by AMBEX Concrete Technologies (Laval, QC, Canada). Table 1 provides the binders’ mineralogical composition in detail.

#### 2.1.3. Sand

In this study, normalized and graded sand, provided by GENEQ Inc. from Ottawa, ON, Canada with a fineness modulus of 2.2-3.2 AFS (grain fineness number), was used following the ASTMC778-21 standard [38].

#### 2.1.4. Water

Deionized water was exclusively used in the preparation of the lixiviates and the cement mixes.

### 2.2. Experimental Approaches

#### 2.2.1. Lixiviate Processing

In this study, hemp hurd was first dried to constant mass at 60 °C [20,22]. It was then soaked and stirred vigorously for 2 h at 90 °C in deionized water before the filtration. For extraction processes, the ratio of 5:1 was used for water/hurd. The rotavapor was used to make a powder and then, the lixiviates were made according to the concentrations of 0.5%, 2%, and 5% and cooled to the ambient temperature (23 °C). The pH of 6.7 was measured using a pH Meter (Metrohm 780, Herisau, Switzerland). Table 2 displays the chemical composition of the investigated hemp hurd in this study and the literature. Figure 2 shows the preparation process of the lixiviates and Figure A1 in Appendix A show the process of hot water extraction. 

#### 2.2.2. Preparation of Cement Mortars and Paste

In this study, the cement, lixiviate, and sand were mixed to make mortar, whereas cement pastes were produced only from cement, water and lixiviate. An amount of 3 g of cement was manually mixed with water and lixiviate for 2 min to prepare a paste using a stainless-steel spatula [40]. The cement pastes were mixed according to the proportions in Table 3.

Normalized Ottawa sand was used for mortar mixtures with a mixing process following ASTM C305-20 standard. It should be noted that 700 g of cement and 1050 g of sand were used in accordance with ASTM C305-20 standard [41]. Table 4 details the composition of cement mortars.

#### 2.2.3. Calorimetry Measurements

The hydration of cement paste was determined using isothermal calorimetry analysis (TA Instruments, TAM AIR 8, New Castle, DE, USA) at 23 °C with a reference sample of Fontainebleau sand. This analysis recorded the heat flux and total amount of heat released during 7 days of measurements [18,29]. The measurements were conducted on cement paste containing hemp hurd lixiviates. For all prepared mixes, the water–cement ratio was kept constant (W/C = 0.5). After mixing, the samples were placed in the calorimeter to observe and determine the cement hydration behaviour. It should be noted that the measurements were carried out three times.

#### 2.2.4. Compression Tests

In this study, compression tests were conducted on specimens made of cement mortars. The moulds of 50 × 50 × 50 mm^3^ were filled to half capacity in order to produce testing samples. Firstly, the half-filled moulds were kept on the shaking table for one minute. After the initial shaking, the moulds were filled again and placed back on the shaking table for one more minute. To ensure that the sample with high lixiviate concentration had reached the final setting and curing times, the samples with 5% lixiviate concentration were, therefore, de-moulded 72 h after casting. In contrast, the final setting times were 48 h and 24 h for the samples with 2% and 0.5% lixiviate concentration, respectively. Therefore, demolding time was calculated based on the concentration of lixiviate. These samples were then cured in water under a humidity chamber at 23 °C. The W/C ratio remained fixed at 0.5 for all types of cement. After the specified curing time, the specimen was removed from the water. Residual water was then wiped out from the surface. The compression tests were performed using the RIEHLE compression/tension Tester (RIEHLE, Baden-Wurttemberg, Germany, Figure A3 in Appendix A). The specimen should be aligned centrally on the base plate of the machine which ensures that the load is applied to the opposite sides of the cube casts. The load was applied gradually at the rate of 2500 lbs/10 s continuously until the specimen failed. The maximum load and the failure types were recorded according to ASTM C203-05 standard [42]. It should be noted that a minimum of 3 specimens were tested to determine the compression strength of mortars.

#### 2.2.5. Thermogravimetric Analysis (TGA)

TGA can evaluate the degree of hydration, and mass loss of different compounds on cement mortars. In the current study, the cube cast of cement mortars after compression tests was crushed to obtain small pieces. The fragments were then crushed using a mortar and pestle until particles with a diameter of approximately 0.5 mm were obtained. To inhibit hydration, water exchange, and carbonation, the samples were kept in methanol for one week. During the storage period, the methanol was changed at least three times to prevent hydration. Before testing, the specimens were dried in a desiccator for about a week [43]. Approximately 25 mg of each sample was heated from 21 °C to 995 °C at a rate of 10 °C/min under nitrogen using TGA Dupont 2000 (Figure A4 in Appendix A).

#### 2.2.6. X-ray Powder Diffraction (XRD)

XRD samples were prepared by cutting the aged samples into appropriate slices for the XRD sample holder. Cutting was performed using a low speed saw to guarantee that no hydrate phases were destroyed by preparation. The samples were measured in a diffractometer X’Pert pro-MPD from the PANalytical company (Worcestershire, UK, Figure A5 in Appendix A). Applying 40 KV and 40 mA, samples were measured from 5 to 70 °C with a step size of 0.02 °C and a counting rate of 0.5 s/step. Evaluation of samples was performed by applying the Rietveld method combined with an external standard method.

#### 2.2.7. Scanning Electron Microscopy (SEM)

The microstructure of binders at 12 h and 7 days was evaluated with a Phenom XL from Thermofisher Company Image (Waltham, MA, USA, Figure A6 in Appendix A), with a backscattered electron (BSE) detector, performed at an accelerating voltage of 15 kv. Prior to evaluation, the hydration of the sample was stopped by solvent exchange with isopropanol and vacuum-impregnated with low-viscosity epoxy resin, polished which diamond discs of 220–1 µm at 200 rpm using ethanol as a lubricant. Polished samples were not coated, but others that had not been polished were coated using carbon with a thickness of 20 nm.

## 3. Result and Discussion

### 3.1. Effect of Lixiviates on Hydration Kinetics of GU and CSA Cement Pastes

In this section, isothermal calorimetry measurements were conducted to assess the effect of Hemp hurds lixiviates on the hydration of GU cement and CSA cement. In other words, the effect of the concentration of lixiviates obtained from hemp hurds on the hydration process of GU and CSA was examined. The effect of hemp hurds lixiviate on the hydration reactions of the cement pastes was investigated by measuring the heat flow released with respect to time. Based on the calorimetric analysis, the heat flow evolution and released heat of the GU and CSA cement pastes with and without lixiviates are presented in Figure 3 and Figure 4. The shift in the position of heat flow peaks can reveal alterations in the kinetics of the hydration process, which in this case indicates a retardation effect. Furthermore, the sum of accumulated heat is attributed to the extent of hydration within the cement paste mix systems. In the first few hours of hydration, a drop in heat release was detected for all samples containing lixiviate concentrations compared to the reference samples (which did not contain lixiviate). This indicated that lixiviates caused a lower amount of heat to be given off by cement pastes. The presence of lixiviate affected the hydration of the cement pastes. However, the degree of hydration varied according to the concentration of lixiviate in the mix. The chronology of hydration, curing and hardening of the GU cement paste is divided into five stages. Stage 1 involves the initial dissolution of cement particles and the formation of ettringite. This is followed by Stage 2, corresponding to the induction period (aka dormant period) where the concentration of calcium ions (Ca^2+^) increases. Stages 3 and 4 correspond to the periods of acceleration and deceleration, during which a significant amount of hydration takes place, resulting in the formation of important compounds such as C-S-H and CH, representing the hydration of C_3_S and C_3_A. Stage 5 entails the diffusion control reaction, where the rest of the hydrated product is formed representing the hydration of C_2_S. Figure 3 depicts the heat flow and total released heat of GU cement with and without lixiviate.

Regardless of cement type, the hydration kinetics slowed when lixiviate was added to the cement paste. Within the first few minutes of mixing, both cement pastes (i.e., GU and CSA) underwent the solubilization reactions, before the heat flow stabilized. The main hydration peaks were after 4 h and 57 min in the reference GU and CSA cement pastes, respectively. Then an increase in the rate of heat evolution occurs which is due to the reaction of alite (C_3_S) with water in the case of GU cement [44] and ye’elimite (C_4_A_3_Š) with water in the case of CSA cement [37]. However, in samples containing 0.5%, 2% and 5% concentrations of hemp hurds lixiviate, the main hydration peaks were delayed. In the case of GU cement paste, the peak of hydration for C_3_S emerged after 3 h for the sample containing 0.5% of lixiviate and 67 h and 91 h for the samples containing 2% and 5% of lixiviate, respectively. Similarly, in the case of CSA cement paste the peak of C_4_A_3_Š hydration occurred after 1 h, 2 h and 2.5 h, for the samples containing 0.5%, 2%, and 5% of lixiviate, respectively. The results showed that the presence of hemp hurds lixiviate had a greater retarding effect on the hydration kinetics of GU cement pastes at concentrations of 2% and 5%. The principal reason is that alite/C_3_S, which accounts for 50–80% of Portland cement and governs the development of properties [45], is more susceptible to the effect of inhibitory substances (sugars, lignin, phenolic compound) than C_4_A_3_Š/ye’elimite (50–92% of CSA) [37]. Figure 3a shows extended dormant periods compared to the reference sample. Mixes with 0.5%, 2% and 5% lixiviate showed extended dormant periods compared to the reference sample (Figure 3a). This phenomenon occurred due to the adsorption of saccharides on anhydrous cement grains in samples that included hemp hurds lixiviate at high concentrations (i.e., 2% and 5%). Kochova et al. [29] studied the effect of Bagasse, coir, hemp, oil palm empty fruit bunch and water hyacinth lixiviates on the hydration kinetics of ordinary Portland cement CEM I 52.5R. Their results showed the retardation effect of more than 3 days with bagasse lixiviate (5:1 water/fibre ratio) without any cement hydration peak. However, lixiviate of coir and hemp had a lower effect on the cement hydration (approximately one hour) of the retardation. Delannoy et al. [20] also studied the influence of three types of shivs on cement hydration. Their results showed that hemp shiv powder had the ability to delay or totally prevent the mixture from setting. When the retarding agents in the lixiviate (sucrose, mannose, lignin, phenolic acids, ash, etc.) were entirely mixed and absorbed during aluminates phases hydration (C_3_A and C_4_AF), the hydration process continued according to its typical pattern with the appearance of C_3_S peak [20]. In the case of GU cement, the excess of lixiviate (i.e., 5%) caused an over-adsorption of saccharides on anhydrous cement grain. This created a thick layer that prevented water from reaching the grains and that halted the reactions [18,20,46]. However, for the CSA cement pastes, the retarding effect was negligible, as indicated in Figure 4. This is primarily because the CSA cement paste exhibits a significantly shorter dormant period compared to the GU ones (Figure 3). In the case of CSA cement, the main part of the hydration of the reference samples (ye’elimite), took place and finished on day one, whereas in the case of GU cement paste the main hydration reactions happened between 7 and 28 days after mixing [36] This is because the hydration reactions of ye’elimite are minimally influenced by the sugars released from the lixiviate [44]. The process of ye’elimite hydration involves dissolution and precipitation, which ultimately results mainly in the formation of ettringite and aluminium hydrates. The leached sugars are not harmful enough to substantially retard the completion of these reactions.

The values of the heat release that are presented in Figure 3b and Figure 4b converge to a value of 225 J/g and 150 J/g for the GU and CSA reference cement pastes, respectively. The heat release revealed a value of 225 J/g for samples that had a concentration of 0.5%, but it dropped to a value of 162 J/g and 158 J/g for samples that included concentrations of 2% and 5% with GU cement, respectively. These results indicate that the heat release decreased with increasing concentration of lixiviate. In contrast to GU cement, CSA cement had nearly the same amount of released heat emitted per gramme of cement. This value was approximately 155 J/g. It is probable that the speed at which ye’elimite hydrated in the first five minutes of the hydration process compared to C_3_S could explain this phenomenon. Table 5 shows the result of heat and total released heat of GU and CSA cement during the 7 days of hydration.

When compared to CSA cement, the hydration kinetics of GU cement exhibited a significant slowdown when exposed to hemp hurd lixiviate. Furthermore, this delay was further amplified with higher concentrations of lixiviate.

### 3.2. Effect of Lixiviates on Hydration Products of GU and CSA Cement Pastes

#### 3.2.1. X-ray Diffraction Analyses

XRD analysis gives detailed information on the hydration process within 12 h and 7 days. Results given in Figure 5 and Figure 6 show the XRD patterns of GU and CSA cements with and without lixiviate, respectively. Ettringite (AFt), portlandite (CH), and calcium carbonate (Cc) are the main hydrated products of GU cement after 12 h of hydration (Figure 5a) for reference one and mix with 0.5% hemp hurds lixiviate. However, there is no hydration peak corresponding to portlandite (CH) on the mixes with 2% and 5% concentration, and the peaks of AFt were nearly similar to the reference samples. The results showed that increasing the concentration of hemp hurds lixiviate in GU cement (i.e., 2% and 5%) delayed the hydration of alite/C_3_S and belite/C_2_S but did not affect the hydration of aluminates (C_3_A and C_4_AF). There are two principal reasons for this phenomenon: (i) firstly, the migration of inhibitory substances (such as sucrose, mannose, lignin, and phenolic acids), as a result of the alkaline hydrolysis of hemicellulose and other substances (starches and lipids), could result in the formation of calcium salts of lignin, polysaccharides, and sugars [30], which interferes with the cement hydration, forms different hydrated products and alters crystalline structures; (ii) secondly, because of the absorption phenomena, potassium (K^+^) and calcium (Ca^2+^) ions could be adsorbed to some extent by the biomolecules released from the lixiviate. Thus, calcium, which is essential for the formation of hydration products and promotes the strength development of cement paste, could be removed from the solution. For these reasons, the formation of hydration products was slowed down by poisoning the calcium nucleation sites [33]. It was indicated in Smith et al.’s [47] study, that the preferential adsorption of saccharide species at initially non-hydrated tricalcium silicate surface sites leads to the delayed formation of calcium silicate hydrates and consequently more significant inhibition of cement hydration. Figure 5b shows that after 7 days of hydration, mixed with 2% hemp hurds lixiviate had a higher amount of CH compared to 12 h, which indicated the growth of the hydration product. Samples with a higher concentration of hemp hurds lixiviate tended to have higher calcium carbonate hydrated. The principal reason is that the Hemp hurd lixiviate undergoes alkaline degradation, resulting in the generation of compounds that can induce the release of carbon dioxide [3,48]. This released carbon dioxide then facilitates the carbonation of CH. As a result, the paste’s Cc experiences an increase. The observed increase in Cc and decrease in CH within the GU cement provide strong evidence supporting the validity of the aforementioned phenomenon. As the amount of hemp hurds lixiviate in the cement paste increased, the formation of CH was inhibited and a notable increase in the presence of non-hydrated products such as C_3_S was detected. Due to the low hydration of C_3_S in the presence of inhibiting substances, gypsum was eventually depleted in the solution due to the rapid hydration of the aluminate’s phases (C_3_A, C_4_AF). This resulted in the formation of AFm through the ions exchange process. The hydration product exhibited continual growth, which correlated with a decrease in the intensity of the C_3_S peak after 7 days of hydration at concentrations of 0.5% and 2% hemp hurds lixiviate. C-S-H seeding may have been inhibited by a competitive interaction with inhibitory substances present in the lixiviate, leading to a decrease in the degree of C_3_S hydration at a concentration of 5% lixiviate [20]. Figure 6a,b depicts the diffractograms of CSA cement hydration product after 12 h and 7 days of hydration, respectively. Ettringite and monosulfoaluminate were the main hydrated products. Peak values for AFt in the pastes containing hemp hurds lixiviate were identical to those in the reference paste, demonstrating that hemp hurds lixiviate had a minimal effect on the formation of AFt. A similar result was observed for the formation of ettringite in the GU cement pastes. The primary reason is that ye’elimite (C_4_A_3_Š), the predominant hydration product in CSA cement [36], reacts with the water and sugars similarly to how aluminates (C_3_A and C_4_AF) in GU cement react with them, resulting in the formation of AFt. After 7 days of hydration (Figure 6b), the mixtures containing 2% and 5% hemp hurds lixiviate exhibited noticeable distinctions. By the same token that the carbonation of CH occurs in the presence of hemp hurd lixiviate in GU cement, the release of carbon dioxide in CSA cement can also induce the carbonation of ettringite. This process leads to the formation of calcite, gypsum, gibbsite, and hydroxy alumina [32,48,49], depending on the reaction:Ca_6_Al_2_(SO_4_)_3_(OH)_12_·26H_2_O + 3CO_2_ → 3CaCO_3_ + 3CaSO_4_·2H_2_O + 2Al(OH)_3_ + 23H_2_O(1)

Grounds et al. [48] observed the same phenomenon in the stability of synthetic ettringite in the presence of CO_2_ under dry and humid conditions. Zhou et al. [32] have also studied the carbonation of pure ettringite, but this time at different temperatures and relative humidities. The crystals have also been studied in powder or compressed pellet form. Zhou affirms that the nucleation of decomposition products is a slow process whose induction period decreases with an increase in temperature or relative humidity.

As discussed earlier, gypsum in the solid phase was used up with the increase of hemp hurd lixiviate in GU cement. As a result, the decrease in sulphate ion concentration caused the migration of alumina (AlO_2_^−^), calcium (Ca^2+^), and hydroxyl (OH^−^) ions toward the periphery of AFt, triggering the transformation reactions [50]. In the case of CSA cement, the conversion of AFt according to Equation 1 was seen, which resulted in the recrystallization of AFt [32]. Is was most likely caused by the absorption of moisture into the pore spaces. However, in the case of GU cement, AFt was enveloped by a C-S-H (calcium-silicate-hydrate) layer. As a result, these ions must have diffused through the C-S-H gel layer to react with the AFt, resulting in a slower transformation process.

#### 3.2.2. Thermogravimetric Analysis (TG/DTG)

TGA/DTG results of hydrated cement paste (Figure 7 and Figure 8) showed the weight loss of hydrated products for the GU and CSA cement pastes due to thermal degradation, respectively. The GU cement paste (Figure 7) showed a progressive weight loss between 50–150 °C which is attributed to the thermal degradation of C-S-H as the main hydrated phase in the Portland cement paste. In the same temperature range as C-S-H, ettringite exhibits mass loss, although, as discussed in the XRD section, it appears to be less severely affected by hemp hurds lixiviate. The weight losses between 400–500 °C and 700–800 °C are attributed to the volatilization of portlandite (CH) and calcium carbonate (CaCO_3_), respectively. In the case of CSA cement, Figure 8 shows for all the samples a progressive weight loss between 50–200 °C attributed to ettringite (as the main hydrated product). This is followed by the thermal degradation of alumina hydroxide and monosulfoaluminate occurring at 200–300 °C and 650–730 °C as minor products. The weight loss between 600 and 700 °C is attributed to the thermal decomposition of calcite in CSA cement.

Compared to GU cement (Figure 7), it appears that CSA cement is less affected by hemp hurds lixiviate. The weight loss of AFt (Figure 8) as the main hydrated product of CSA cement for the sample containing 0.5%, 2% and 5% was nearly similar to the reference sample. In contrast, the weight loss of the main hydrated product of GU cement (i.e., C-H-S) for the sample containing 0.5%, 2% and 5% was smaller compared to the reference one. As explained before, inhibitory substances in the lixiviate, such as sugars, lignin and phenolic acids, can be responsible for this phenomenon. As confirmed by XRD results, the weight loss of calcium carbonate (Cc) went up with the lixiviate content after 12 h and 7 days of hydration. Except for Cc, all the weight loss of hydrated products was under the reference one after 12 h of hydration for GU cement. However, after 7 days of hydration, the samples containing 0.5% and 2% displayed an increase in the weight loss of AFt and CH compared to the reference sample. This is due to the evolution of hydration products when the amount of retarding agents (i.e., sugar, lignin and phenolic acids) was exhausted due to the absorption during the hydration of aluminate phases. The XRD analyses confirmed these results, as the intensity of the C_3_S peaks decreased after 7 days of hydration. GU cement showed weight loss for the AFm phase after 7 days of hydration (Figure 7b) due to the AFt to AFm transformation. CSA cement showed weight loss for the AFm and calcite phases after 12 h and 7 days of hydration (Figure 8a,b) due to the carbonation and recrystallization of AFt. The mechanism of transformation has been described in detail in the XRD section. Furthermore, calcite weight loss in CSA cement can be attributed to the reduction–oxidation reaction of SO_4_^2−^ in the presence of the organic carbon of citric acid. Similar results were observed by Hu et al. [31] who studied the influence of citric acid on the hydration of CSA cement. Table 6 shows the weight loss of different hydrated products of GU and CSA cement after 12 h and 7 days of hydration with and without lixiviate.

#### 3.2.3. SEM Analysis

The microstructure alterations in hydrated GU and CSA cement caused by the addition of 0.5%, 2%, and 5% hemp hurds lixiviate were examined using SEM analysis. The corresponding micrographs are presented in Figure 9, Figure 10 and Figure 11, respectively. The microscopic morphology of the hydrated GU and CSA binders was explored at both 12 h and 7 days, revealing the influence of hemp hurds lixiviate on the microscopic structure of both binders. Crystals of ettringite were identified in the SEM micrographs of GU cement mixed pastes with distilled water, 0.5% and 2% hemp hurds lixiviate after 12 h of hydration (Figure 11(a1–a3)), but the crystals of ettringite were quite tiny at the concentration of 2% (Figure 9(a3)) compared to 0.5% and reference one for GU cement (Figure 9(a1,a2)). Those ettringite crystals were more visible after 7 days of hydration (Figure 9(b1,b2)). Furthermore, at 2% (Figure 9(b3)), a small amount of ettringite was initially detected in the sample before it started transitioning into AFm. As explained previously, the reason is that the gypsum has been completely exhausted in the solution. Then the ettringite crystals formed first, and then AlO_2_^−^, Ca^2+^, and OH^−^ ions reacted with them to initiate the transition. In the case of GU cement, the AFm phase was detected by XRD and DTG in mixes containing significant amounts of hemp hurds lixiviate (i.e., 2% and 5%). For the pastes with the concentration of 5% lixiviate, ettringite was not visible after 12 h of hydration for GU cement, indicating the inhibition of hydration products. However, after 7 days of hydration, syngenite was visible in the sample (Figure 9(b4)). Pantawee et al. [51] indicated that raw hemp hurds had higher potassium oxide (K_2_O) content and could be leached with water. Potassium hydroxide can readily react with calcium sulphate (CaSO_4_) in the hydration process of Portland cement. Then, potassium sulphate (K_2_SO_4_) can accelerate the setting time of gypsum (CaSO_4_·2H_2_O) to form syngenite [20]. In the case of CSA cement, ettringite was present across the board (Figure 10(a1–a4)) and was observed to rise when hemp hurds lixiviate concentration rose, as indicated in Table 6, after 12 h of hydration. This indicated that hemp hurds lixiviate did not affect the formation of ettringite at a young age for CSA cement. After 7 days of hydration, the crystals of ettringite were visible within the pores of CSA cement samples, and a layer of hydrated products was developed over the ettringite crystals for samples containing hemp hurds lixiviate (Figure 10(b2–b4)). As explained before, this was due to the carbonation of ettringite in the presence of carbon dioxide. It is suggested that the first stage corresponds to CO_2_ gas diffusion along the channels, interference with the charge balance at the column surface and competition between SO_4_^2−^ and CO_3_^2−^ ions. This could cause the ettringite crystallites to split and initiate further reaction at defects in the column structure. The second stage would then involve more rapid decomposition as the reaction interface penetrates into and along the column material, disrupting the main structure with rapid formation of calcium sulphate and calcium carbonate [32,49]. As the concentrations of hemp hurds lixiviate increased in the sample, this pattern was observed to intensify correspondingly. At 5% concentration after 7 days of hydration (Figure 10(b4)), crystal ettringite was no longer visible on the surface of the sample, probably due to the total transformation of ettringite on the surface into calcite, gypsum, and alumina hydroxide, according to Equation (1). A rectangular prismatic morphology emerged in the samples during the AFt to AFm phase transformation (Figure 11), probably due to the recrystallization of AFt. Another explanation for the detection of AFm could be the condensation of water in the pores and inter-crystalline space in pellet samples [52]. The microstructural investigation by SEM revealed identifiable phase assemblages, including AFm, AFt, C_2_S, and the remaining reactants, in CSA with 5% hemp hurds lixiviate after 7 days of hydration.

This study focuses exclusively on investigating the impact of hemp hurds lixiviate on the hydration process of ternary binders, specifically CSA and GU cement. However, certain aspects, such as the influence of carbohydrates such as starches and lipids (oils, waxes, etc.) on hydration kinetics and the formation of distinct hydrated products, remain unclear. Exploring these factors will be the focus of our future research endeavours and investments.

### 3.3. Effect of Lixiviates on Compression Strength of Cement Mortars Made with GU and CSA

The compressive strength results of mortars at 3, 7, and 28 days of hydration are illustrated in Figure 12a,b. The statistical analysis of the data is shown in Table 5. The compressive strength variability is significantly affected by the cement type, lixiviate incorporation, lixiviate concentration, and hydration time. Compressive strength was found to be significantly affected by interactions between the cement type and lixiviate content. The mechanical properties of CSA cement outperform those of GU cement. After a hydration period of 3 days, it was obvious that the CSA samples demonstrated a notable increase in compressive strength, surpassing the reference sample of GU cement by 44.5%. The CSA samples outperformed the GU samples in mechanical performance by 36.7%, 63.5%, and 71% for samples containing concentrations of 0.5%, 2%, and 5%, respectively. As mentioned earlier, although the hydration process of ye’elimite remained affected by the presence of lixiviate, the biomolecules released by the lixiviate had a greater negative impact on C_3_S.

The addition of a 5% concentration resulted in a significant reduction in the 28-day compressive strength of both GU and CSA. The strength decreased by up to 69.3% for GU cement and up to 62.9% for CSA cement, compared to the reference sample. As observed in the isothermal calorimetry results, the retarding effect of all admixtures was more pronounced at a concentration of 5%, impacting the compressive strength between 3, 7 and 28 days of hydration. Lixiviate decreased compressive strength for all ages at 0.5, 2, and 5% concentrations compared to the reference. For the GU cement mortars, compressive strength was reduced by 6.5%, 58%, and 75% after 3 days; 5.5%, 61.5%, and 70.5% after 7 days; and 4.5%, 66%, and 69% after 28 days, while the CSA cement’s values were reduced by 6.5%, 35%, and 51.5% after 3 days; 13.5%, 44%, and 58.5% after 7 days; and 6%, 38.5%, and 62% after 28 days of hydration. When compared to CSA cement, which lost just 38% of its mechanical properties after curing for 28 days at the concentration of 2% lixiviate, GU cement lost 66%. According to these results, the compressive strength of GU was significantly affected at the concentration of 2% lixiviate, in contrast to CSA cement. The results demonstrate that CSA cement exhibited satisfactory mechanical performance when subjected to 2% lixiviate. It is presumed that untreated hemp hurds would release less than 2% lixiviate when incorporated into cement pastes. Given these circumstances, CSA cement presents itself as a superior choice for applications involving hemp hurds as a bio-aggregate, serving as an optimal binder option. Since AFt is the primary hydrate product during CSA hydration, the carbonation of AFt to calcite, gypsum, and aluminum hydroxide reduces the volume occupied by the cementitious matrix, thus increasing porosity, and reducing mechanical properties. This may be one of the causes of the drop in the mechanical properties. As explained before, the low mass loss of hydrated mortars shown in XRD and DTG results in the presence of lixiviate are closely linked to the low mechanical properties of cement mortars.

## 4. Conclusions

The objective of this study was to investigate the physio-mechanical properties of GU and CSA in relation to their suitability for applications involving bio-aggregate such as hemp hurds. This study aimed to determine which binder would be the most suitable for the development of hemp hurds incorporated into cementitious composites. The principal findings are presented as follows:Due to incomplete hydration of the cement pastes, it was found in the experiment that the presence of the lixiviate obtained from hemp hurds slowed down the setting of the binders. At a high concentration of 5% lixiviate, it was observed that the setting of the GU paste was completely inhibited. Furthermore, an increase in the percentage of lixiviate from 0.5% to 5% in the GU cement paste resulted in a shift of the C_3_S hydration peak to later times. Specifically, when the GU mixture contained a concentration of 5% lixiviate, the peak of C_3_S hydration was observed after 4 days, and only the hydration of aluminates was evident. In the case of CSA cement, which primarily consists of ye’elimite, the addition of hemp hurds lixiviate resulted in a rapid hydration reaction, as indicated by calorimetry measurements. It was observed that hemp hurds lixiviate had a more significant impact on the hydration process of GU cement, while CSA cement was relatively less affected.XRD and DTG analyses were employed to assess the nature and quantity of hydrated products. The results of these tests revealed an inverse correlation between the presence of hemp hurds lixiviate and the formation of hydrated products. The absence of portlandite (CH) hydrate at 5% concentration after 7 days of hydration indicates that the hydration of C_3_S was inhibited, leading to poorer GU cement mechanical properties. Ettringite, on the other hand, was present early in the hydration process, revealing that hemp hurds lixiviate had a smaller impact on the hydration process of the aluminate phase, which includes C_3_A and C_4_AF.Finally, the effect of the extractives present in lixiviate on the mechanical properties of GU and CSA was studied by measuring the compression strength of cement mortars. The compression strength decreased as the extractable content in the solution increased. The extractives in lixiviate had a significant impact on the compression strength of GU cement. However, CSA cement developed better mechanical properties compared to GU cement up to the concentration of 2% lixiviate.

This study focused exclusively on investigating the impact of hemp hurd lixiviate on the hydration process of ternary binders, specifically CSA and GU cement. However, certain aspects, such as the influence of carbohydrates such as starches and lipids (oils, waxes, etc.) on hydration kinetics and the formation of distinct hydrated products, remained unclear. Exploring these factors will be the focus of our future research endeavours and investments.

This study revealed that CSA cement was more compatible with hemp hurd than GU cement to produce bio-aggregate reinforced cementitious panels due to the hydration behaviour and hardened state properties.

## Figures and Tables

**Figure 1 materials-16-05561-f001:**
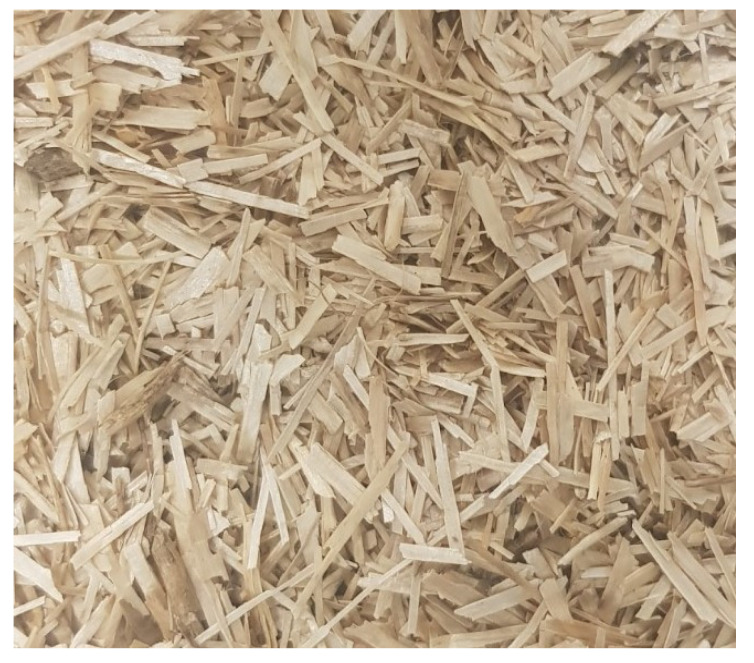
Natural organic hemp hurds.

**Figure 2 materials-16-05561-f002:**
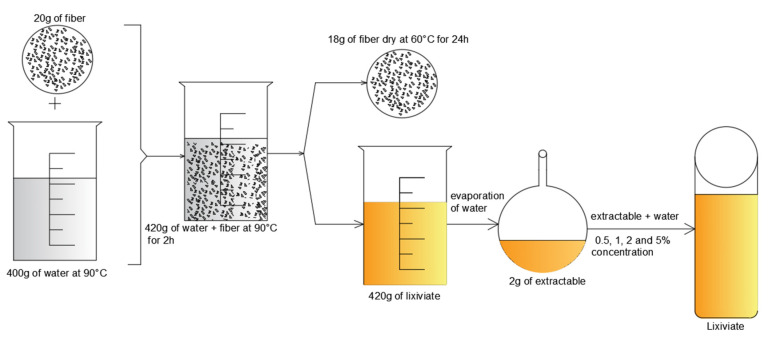
Schematic diagram of the preparation process of the lixiviates.

**Figure 3 materials-16-05561-f003:**
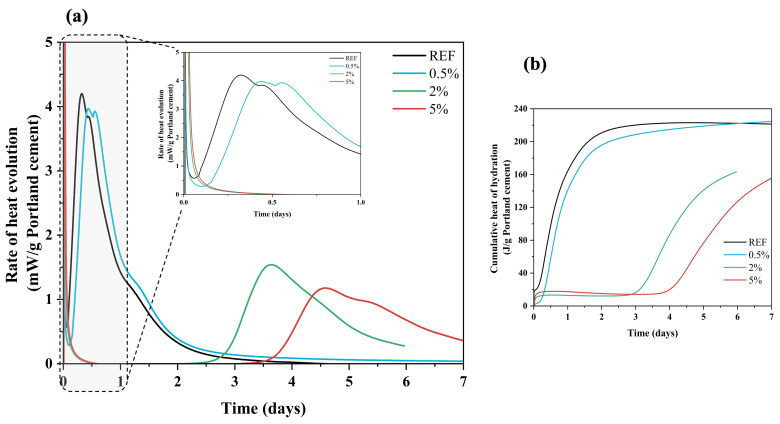
(**a**) Heat flow and (**b**) released heat of GU cement with and without Hf lixiviate. DW: Distilled Water.

**Figure 4 materials-16-05561-f004:**
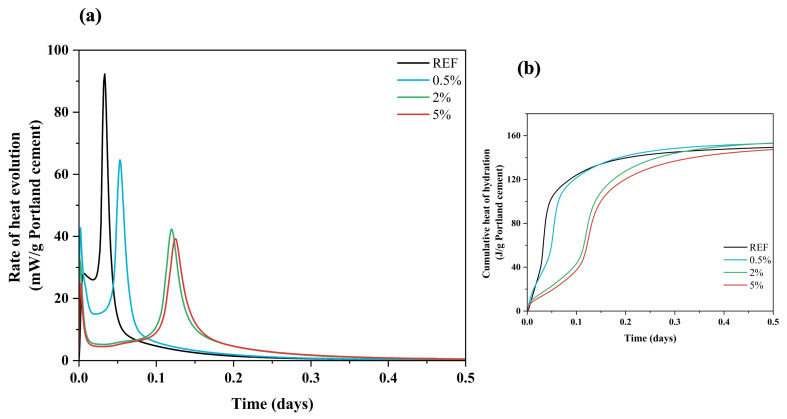
(**a**) heat flow and (**b**) released heat of CSA cement with and without hemp hurd lixiviate. DW: Distilled Water.

**Figure 5 materials-16-05561-f005:**
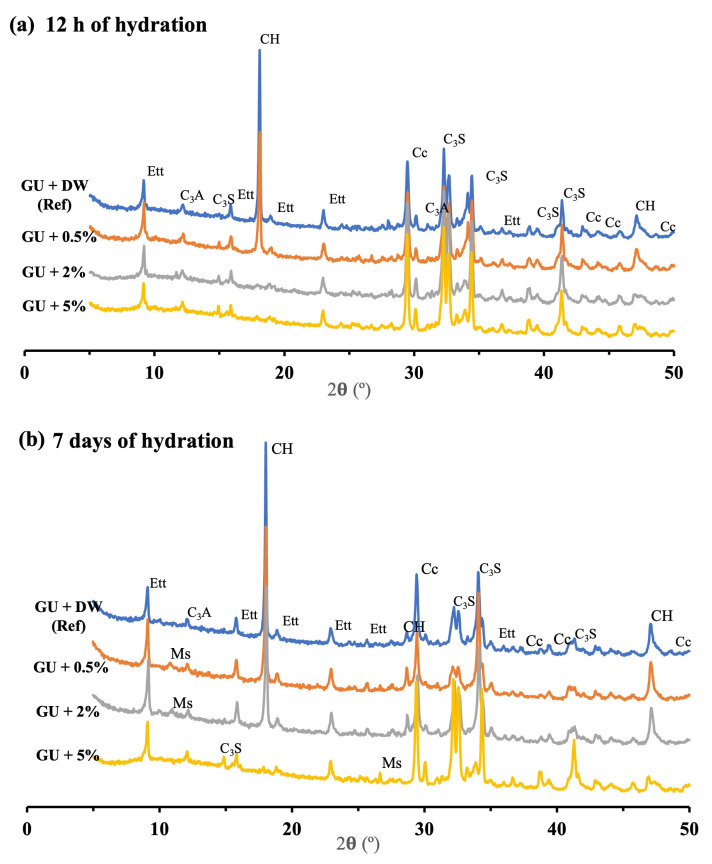
XRD patterns of GU cement with and without lixiviate after 12 h of hydration, AFt: ettringite; Cc: Calcium carbonate; CH: portlandite; Ms: Monosulfoaluminate; C_3_S; silicate tricalcium.

**Figure 6 materials-16-05561-f006:**
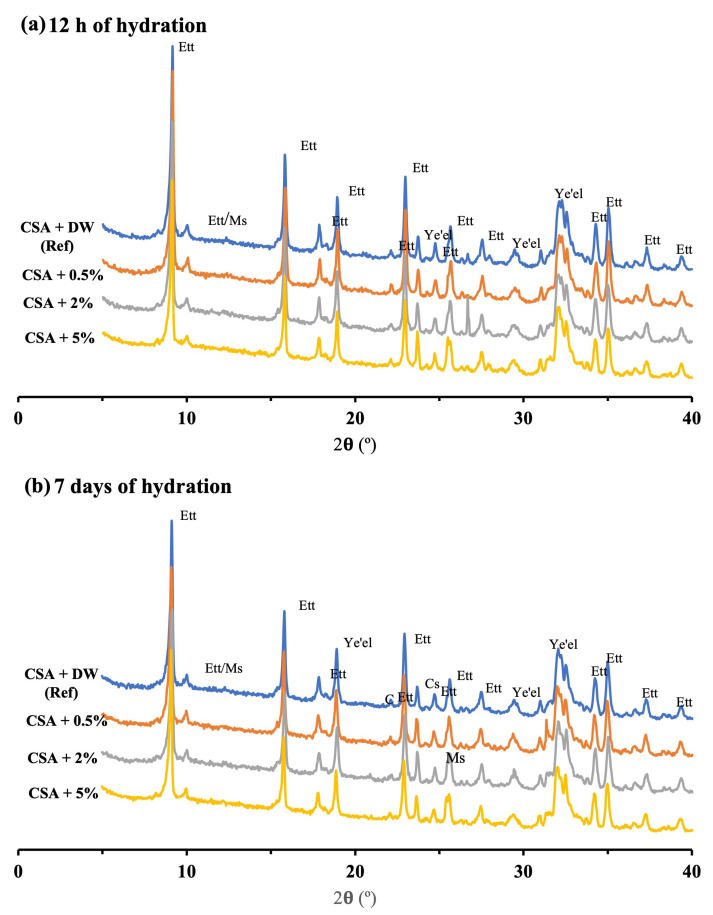
XRD patterns of CSA cement with and without lixiviate. AFt: ettringite; Cc: Calcium carbonate; Ms: Monosulfoaluminate; Ye’el: ye’elimite; Cs: Calcium silicate, B: Belite; G: Gypsum; Al: alumina hydroxide.

**Figure 7 materials-16-05561-f007:**
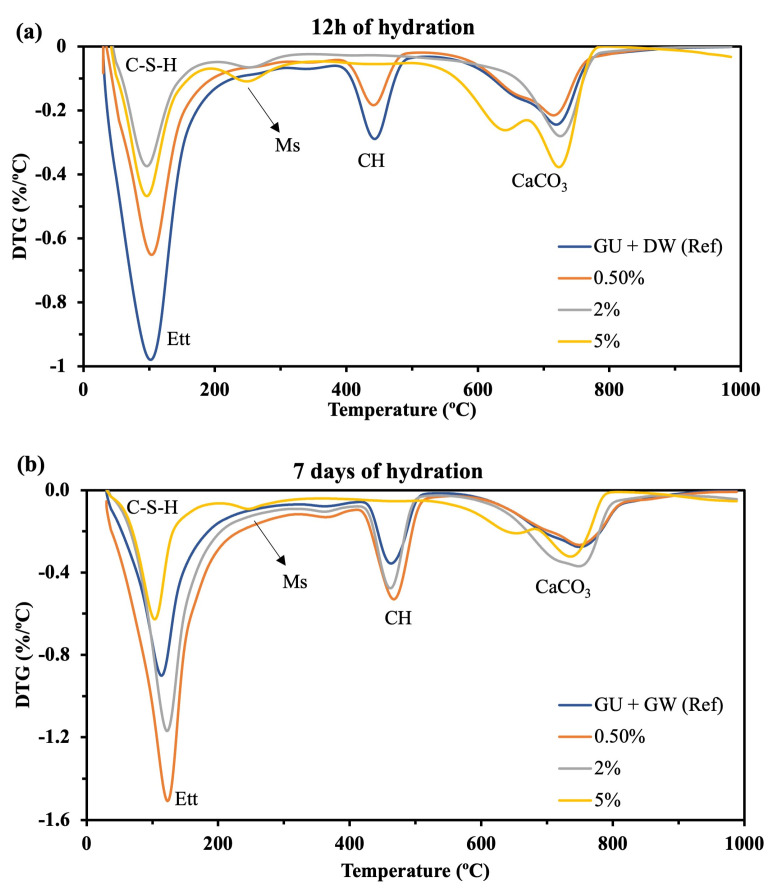
TGA/DTG of GU cement with and without hemp hurd lixiviate after (**a**) 12 h and (**b**) 7 days of hydration (C-S-H: Calcium silicate hydrate; CH: portlandite; CaCO_3_: Calcium carbonate; Et: Ettringite; Ms: Monosulfoaluminate.

**Figure 8 materials-16-05561-f008:**
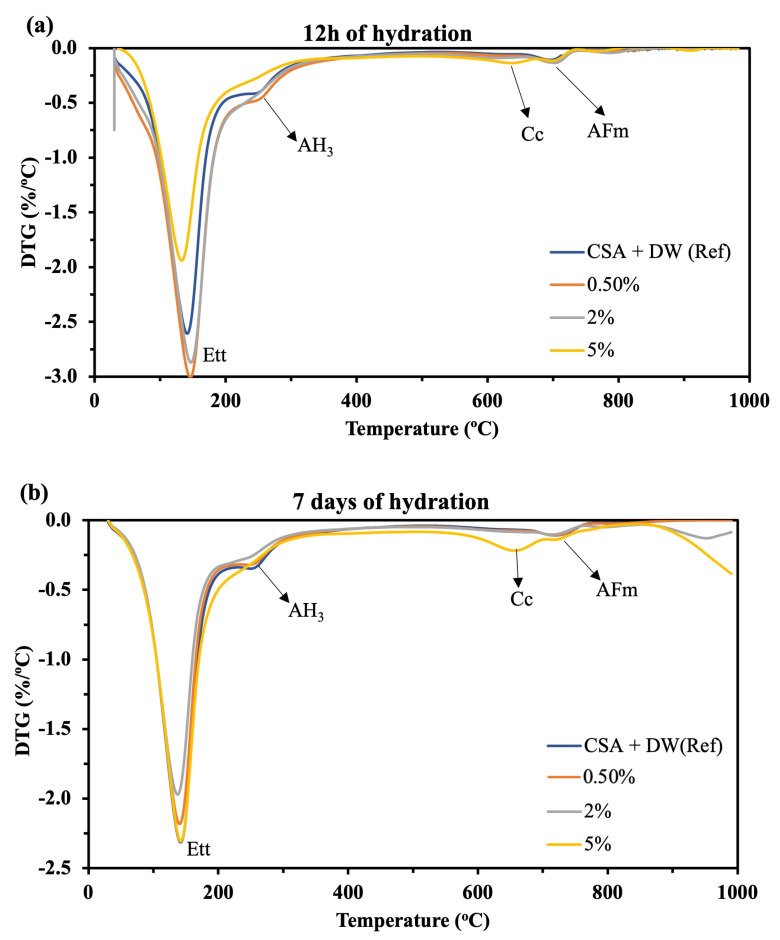
TGA/DTG of CSA cements with and without Hf lixiviate after (**a**) 12 h and (**b**) 7 days of hydration. AFt: Ettringite; AH_3_: Aluminium hydroxide; Cc: Calcite.

**Figure 9 materials-16-05561-f009:**
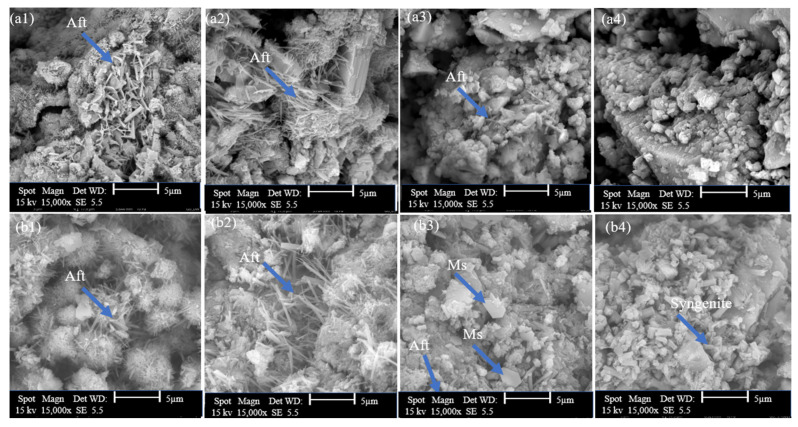
SEM diagrams of GU cement with and without lixiviate (**a**) 12 h of hydration and (**b**) 7 days of hydration: (**a1**,**b1**) with DW, (**a2**,**b2**) 0.5%, (**a3**,**b3**) with 2% and (**a4**,**b4**) with 5% hemp hurds lixiviate.

**Figure 10 materials-16-05561-f010:**
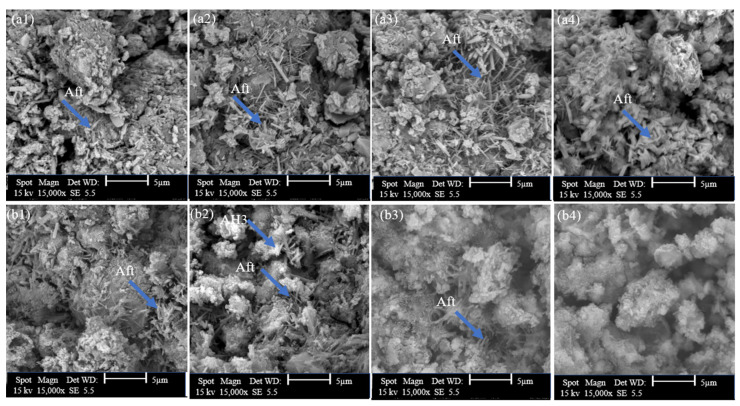
SEM diagrams of CSA cement with and without lixiviate (**a**) 12 h of hydration and (**b**) 7 days of hydration: (**a1**,**b1**) with DW, (**a2**,**b2**) 0.5%, (**a3**,**b3**) with 2% and (**a4**,**b4**) with 5% hemp hurds lixiviate.

**Figure 11 materials-16-05561-f011:**
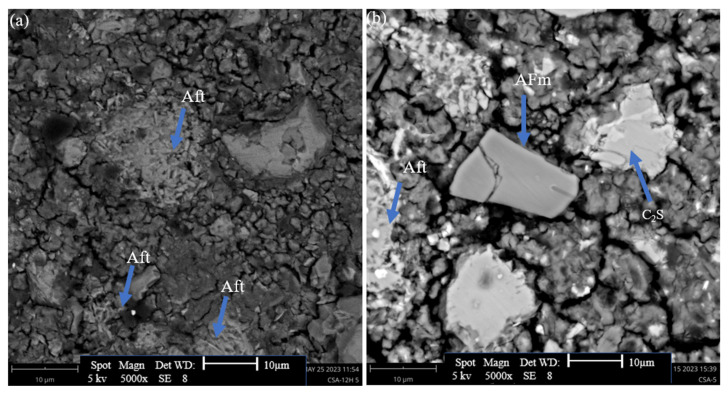
SEM diagrams of CSA cement with 5% hemp hurds lixiviate, with the presence of (**a**) Aft and (**b**) AFm phase after 7 days of hydrations.

**Figure 12 materials-16-05561-f012:**
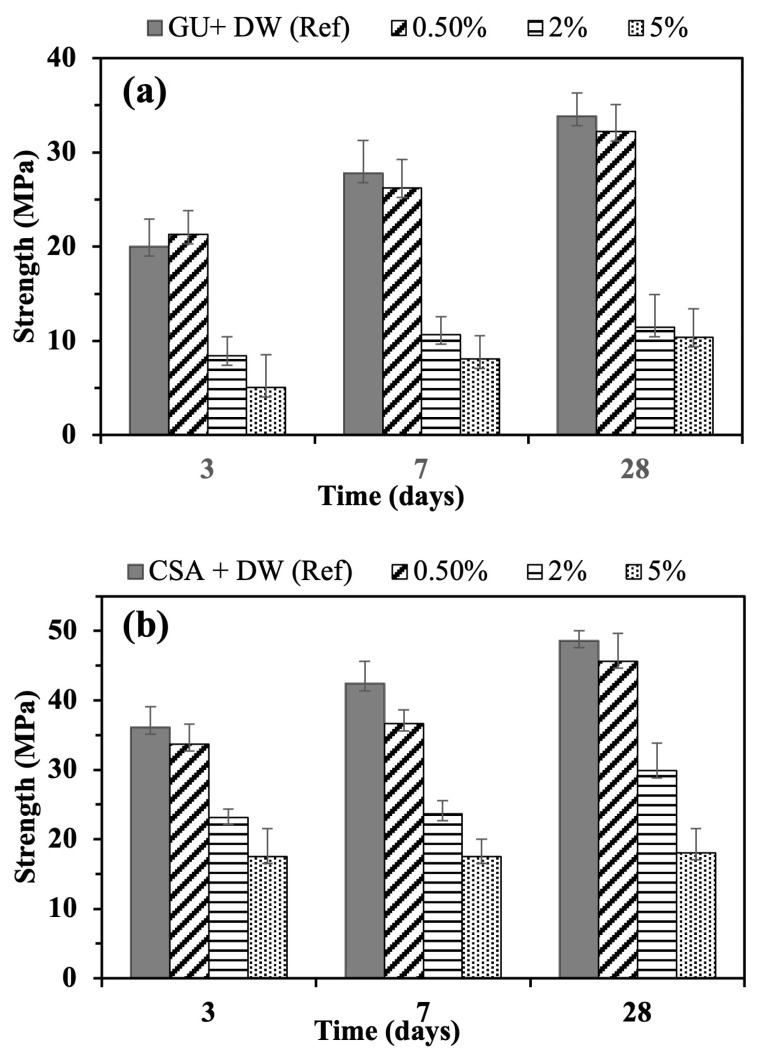
Compression strength of (**a**) GU, and (**b**) CSA cements, with and without hemp hurds lixiviate.

**Table 1 materials-16-05561-t001:** Mineralogical composition of GU and CSA cement (wt.%).

	Content (wt.%)	
Mineral Composition	In GU	In CSA
Tricalcium silicate (C_3_S)	53.7	-
Dicalcium silicate (C_2_S)	17	20.8
Ye’elimite (C_4_A_3_Š)	-	54.3
Tricalcium aluminate (C_3_A)	7.18	-
Tetracalcium aluminoferrite (C_4_AF)	7	9.3
Akermanite (C_2_MS_2_)	-	4.5
Free lime	-	0.6
C2Sα’ high	-	8.3
Oxide de fer (Y-Fe_2_O_3_)	-	1
Ferrite (C_6_AF_2_)	-	1.2

GU cement contains 3.8% gypsum whereas CSA cement contains 15.4%.

**Table 2 materials-16-05561-t002:** Chemical composition of hemp hurd (%) [2,15,16,22,39].

Component	“Authors Analysis” %	Literature
Cellulose	48.3	(45.6–49.2)
Hemicellulose	28	(17.8–21)
Lignin	18	(4–17.2)
Pectin	0.7	(0.4–19.04)
Others	5	(1.9–3.1)

**Table 3 materials-16-05561-t003:** Mix proportions of cement pastes.

Mixtures	Water Mass (g)	Extractable Compound (g)	Lixiviate Mass (g)	Lixiviate Concentration (%)
Ref	1.512	-	-	-
	1.5	0.0075	1.5075	0.5
	1.5	0.0151	1.5151	1
	1.5	0.0301	1.5301	2
	1.5	0.0781	1.5941	5

**Table 4 materials-16-05561-t004:** Mix proportions of cement mortars.

Mixtures	Water Mass (g)	Extractable Compound (g)	Lixiviate Mass (g)	Lixiviate Concentration (%)
Ref	359	-	-	-
	359	1.8	360.80	0.5
	359	3.63	362.63	1
	359	7.33	366.33	2
	359	18.89	377.89	5

**Table 5 materials-16-05561-t005:** Isothermal calorimeter result.

Type of Cement	Lixiviate Concentration (%)	Main Heat Flow Peak (mW/g)	Occurrence of Main Heat Flow Peak (h)	Cumulative Heat 7 Days (J/g)
GU	DW (Ref)	0.012	2	225
	0.5	0.0115	3	225
	2	0.0045	67	162
	5	0.0035	91	158
CSA	DW (Ref)	0.275	0.08	150
	0.5	0.18	1.5	152.5
	2	0.125	2.5	152.5
	5	0.12	3	150

**Table 6 materials-16-05561-t006:** Weight loss of different hydrated compounds.

Hydrated Product	GU (12 h)				CSA (12 h)			
	DW	0.5%	2%	5%	DW	0.5%	2%	5%
CH (g)	1.18	0.71	0	0	-	-	-	-
CaCO_3_ (g)	1.36	1.51	1.93	4.33	-	-	-	-
AFm (g)	0	0	0.15	0.24	0	0.33	0.35	0.22
AFt (g)	-	-	-	-	11.78	12.48	12.06	8.98
AH_3_ (g)				-	0.28	0.16	0.06	0.06
Calcite (g)	-	-	-	-	0	0	0	0.26
		GU (7 days)				CSA (7 days)		
CH (g)	1.41	2.22	1.76	0	-	-	-	-
CaCO_3_ (g)	2.89	2.31	3.36	3.29	-	-	-	-
AFm (g)	0	0	0.14	0.22	0.269	0.266	0.13	0.05
AFt (g)	-	-	-	-	11	10.1	8.21	11.11
AH_3_ (g)	-	-	-	-	0.52	0.28	0.023	0.03
Calcite (g)	-	-	-	-	-	-	-	0.65

## Data Availability

No new data were created or analyzed in this study. Data sharing is not applicable to this article.

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
