# Peer review of "Retarding Effect of Hemp Hurd Lixiviates on the Hydration of Hydraulic and CSA Cements"

_materials, 2023, doi:10.3390/ma16165561_

Round 1
Reviewer 1 Report
Overview report-2522943
Executive Summary
The use of wood-fiber boards as environmentally friendly cement composites is currently relevant worldwide. A big environmental problem is the reduction of the carbon footprint from the production of binders.
Therefore, studies aimed at investigating the influence of the woody part of fast growing plants such as hemp on the hydration of binders are of scientific interest.
This allows us to speak about the relevance of the task.
Comments on the general concept
Statement of the problem.
The authors of the article aim to study the effect of bioaggregate lixiviates (hemp haulm) on the hydration kinetics of calcium sulfoaluminate (CSA) cement as a low-carbon alternative to conventional Portland cement (PC).
The authors showed that the extractives in lixiviate had a significant effect on the hydration processes and compressive strength of GU cement. However, CSA cement showed better mechanical properties compared to GU cement up to a concentration of 2% lixiviate.
The "Materials and Methods" section describes in sufficient detail the preparation of samples, and their investigation by means of modern physico-chemical methods of analysis. The results of the study are presented clearly, contribute to the scientific field of study, and are fully and correctly formulated. The conclusions are supported by the analysis of the results and provide answers to the research questions. The used literature corresponds to the stated topic of the study.
Remarks
1. The purpose of the study needs to be more clearly and specifically stated.
2. When describing the XRD and TG data, the authors make references to the works [32. 51.52. 53.54.55.56] on the effect of CO2 on ettringite. From the authors' reasoning it is not clear whether the decomposition of ettringite occurs by the action of hemp hurd lixiviate.
3. If the process of ettringite decomposition is present but depends on the kinetic factor, the question remains about the durability of hemp hurd and calcium sulfoaluminate (CSA) cement products.
4. The numbering of the tables should be harmonized. In the text the reference to table 3 is given (in the absence of table 2), but in fact it is table 2. The same with tables 3 and 4 and so on.
Conclusion
The scientific article is quite informative for specialists and is of interest to researchers dealing with the problem of utilization of woody parts of plants and various types of binding agents.
The article can be recommended for printing with finalization of the remarks made.

Author Response
Comment |
Response/Amended Manuscript (Highlighted in Light Blue) |
The authors of the article aim to study the effect of bioaggregate lixiviates (hemp haulm) on the hydration kinetics of calcium sulfoaluminate (CSA) cement as a low-carbon alternative to conventional Portland cement (PC). The authors showed that the extractive in lixiviate hat a significant effect on the hydration processes and compressive strength of GU cement. however, CSA cement showed better mechanical properties compared to GU cement up to a concentration of 2% lixiviate. The ‘Materials and Methods’ section describes in the sufficient detail the preparation of samples, and their investigation by means of modern physico-chemical methods of analysis. The results of the study are presented clearly, contribute to the scientific field of study, and are fully and correctly formulated. The conclusions are supported by the analysis of the results and provide answers to the research questions. The used literature corresponds to the stated topic of the study. |
The authors appreciate the comment and feedback.
|
1- ’ The purpose of the study needs to be more clearly and specifically stated’ |
The authors appreciate the comment. To clarify this matter, the following sentences were added to the introduction on page 7, line 4 as:
“The primary goal of this study is to investigate the adverse effects of hemp hurd leachates on the hydration process of CSA, which has the potential to serve as a more environmentally friendly substitute for OPC in WWCB applications. The research aims to quantify the extent to which these negative effects can delay the setting time and reduce the mechanical properties of cementitious wooden panels with respect to the concentration of hemp hurd leachates. Furthermore, this study intends to provide valuable insights to subsequent studies to optimize the pre-treatment methods necessary to ensure the compatibility of hemp hurd with the binders used in the production of WWCB.” |
2- ’When describing the XRD and TG data, the authors make references to the works [32. 51 .52 53. 54. 55. 56] on the effect of CO2 on ettringite. From the authors’ reasoning it is not clear whether the decomposition of ettringite occurs by the action of hemp hurd lixiviate’ |
The authors appreciate the comment. Discussion of this point is amended and revised below: (Page 20, line 3) “the Hemp hurd lixiviate undergoes alkaline degradation, resulting in the generation of compounds that can induce the release of carbon dioxide. This released carbon dioxide then facilitates the carbonation of CH. As a result, the paste's Cc experiences an increase.”
(Page 20, line 6) “The observed increase in Cc and decrease in CH within the GU cement provide strong evidence supporting the validity of the aforementioned phenomenon.”
(Page 21, line2) “By the same token that the carbonation of CH occurs in the presence of hemp hurd lixiviate in GU cement, the release of carbon dioxide in CSA cement can also induce the carbonation of ettringite. This process leads to the formation of calcite, gypsum, gibbsite, and hydroxy alumina.”
|
3- If the process of ettringite decomposition is present but depends on the kinetic factor, the question remains about the durability of hemp hurd and calcium sulfoaluminate (CSA) cement products. |
The authors appreciate the comment and they agree with their insights. They acknowledge the significance of this matter and plan to thoroughly investigate it in future research endeavors.
|
4- The numbering of the tables should be harmonized. In the text the reference to table 3 is given (in the absence of table 2), but in fact it is table 2. The same with tables 3 and 4 and so on. |
The numbering of the tables has been amended to ensure consistency with the text.
|

Reviewer 2 Report
From the reviewer's point of view, I can state that the paper has a classical structure, including 4 main parts, an introduction to the problem, a description of materials and methods, a presentation of results, and including discussion and conclusions. The different parts of the paper are well-balanced. The authors refer to a large number of papers and other materials. In my opinion, English is error-free, and the figures, graphs, and tables used appropriately complement the text of the paper. The results are presented with clear graphs and figures. The assumptions made by the authors and the selected building materials have been chosen expediently.
The testing methods used are correct and appropriate for the problem in terms of proving the defined assumptions. From my point of view, the results obtained are not surprising, they confirm the theoretical assumptions and conclusions from the published literature. At the same time, they complement and extend them. Thus, certainly, the presented conclusions will be very valuable for the professional community in the field of materials engineering. I also appreciate the fact that the authors conclude by providing information on the future direction of the practice. I have no substantial comments on the paper. Given that the use of hemp is a relatively new topic in research related to cement, and given that there are a limited number of studies that address the precise effect of hemp lixiviates on cement hydration, I recommend the editors to publish it without major modifications.
For authors to add:
It is recommended to indicate for which types of structural parts the presented technology is suitable.
The presented technology will contribute to the reduction of emissions, what will be the impact on the price of the final products?
What other types of tests will need to be carried out?
Author Response
Comment |
Response/Amended Manuscript (Highlighted in Light Blue) |
From the reviewer’s point of view, I can state that the paper has a classical structure, including 4 main parts, an introduction to the problem, a description of materials and methods, a presentation of results, and including discussion and conclusions. The different parts of the paper are well-balanced. The authors refer to a large number of papers and other materials. In my opinion, English is error-free, and the figures, graphs, and tables used appropriately complement the text of the authors and the selected building materials have been chosen expediently.
The testing methods used are correct and appropriate for the problem in terms of proving the defined assumptions. From my point of view, the results obtained are not surprising, they confirm the theoretical assumptions and conclusions from the published literature. At the same time, they complement and extend them. Thus, certainly, the presented conclusions will be very valuable for the professional community in the field of materials engineering. I also appreciate the fact that the authors conclude by providing on the future direction of the practice. I have no substantial comments on the paper. Given that the use of hemp is a relatively new topic in research related to cement and given that there are a limited number of studies that address the precise effect off hemp lixiviates on cement hydration, I recommend the editors to publish it without major modifications.
|
The authors appreciate the comment and feedback.
|
1- “It is recommended to indicate for which types of structural parts the presented technology is suitable”.
|
The authors appreciate the comment. The sentence bellow was added on the introduction section on page 7 lines 6.
“The presented technology is suitable for wood-wool cement board, uses in ceiling tiles and insulation wall panels.” |
2- “The presented technology will contribute to the reduction of emissions, what will be the impact on the price of the final products?” |
The authors appreciate the comment. While there might be associated costs with the pre-treatments required to remove hemp hurd extractives before their incorporation into cementitious composites, it is essential to consider the potential enhancement in durability that these treatments can provide. The authors recognize the importance of this aspect, but they consider it a subject of separate and dedicated research in the future. |
3- “What other types of tests will need to be carried out?”
|
The others tests to be carried out are as follows: · Modulus of rupture (MOR) on wool-wood cement boards with hydrothermal treated and untreated hemp shivs and CSA, GU and White cements · Modulus of Elasticity (MOE) on wool-wood cement boards with hydrothermal treated and untreated hemp shivs and CSA, GU, and White cements · Water absorption (WA) · Thickness swelling (Ts) · SEM Images Characterizations.” The authors are in the process of completing these tests and anticipate publishing the results in a separate article soon. |

Reviewer 3 Report
The manuscript addresses the topic of construction materials in terms of sustainable cementitious composites. This research is quite well written and the results support the conclusion of this manuscript. However, the manuscript requires deep revision according to the comments below:
1) Abstract. The abstract need to refer to the obtained qualitative results as well quntitative results. Giving the fitting goodness without a necessary background is not informative. Therefore, it is recommended to correct abstract.
2) Introduction. In the first section there is no in-depth discussion of the behavior of cementitious composites modified by supplementary cementitious materials (SCMs) like fly ash or silica fume. Moreover, synergy between these materials is also very important in regards to reduce carbon footprint. There are recent articles published on this subject that should be discussed and referenced, such as:
- „Compound Effects of Sodium Chloride and Gypsum on the Compressive Strength and Sulfate Resistance of Slag-Based Geopolymer Concrete”, Buildings 2023.
- “Combined effect of coal fly ash (CFA) and nanosilica (nS) on the strength parameters and microstructural properties of eco-friendly concrete”, Energies 2023.
- „Synergetic Effect of Superabsorbent Polymer and CaO-Based Expansive Agent on Mitigating Autogenous Shrinkage of UHPC Matrix, Materials 2023.
3) Expeiments. Please provide photos showing specimens during preprations and testing. Moreover, please provide more details regarding equipments used in the studies.
4) Results. All charts with test results should be presented in colors.
5) SEM studies. SEM images from figs. 9 and 10 are too small. In addition, all photos should have scale and magnification clearly marked.
6) Conclusions. Conclusions should be changed. Conlusions should be presented in several strong bullets.
Minor editing of English language required.
Author Response
Comment |
Response/Amended Manuscript (Highlighted in Light Blue) |
The manuscript addresses the topic of construction materials in terms of sustainable cementitious composites. This research is quite well written, and the results support the conclusion of this manuscript. However, the manuscript requires deep revision according to the comment below. |
The authors sincerely appreciate the comment and feedback received. While we acknowledge the significance of your input, we also believe in maintaining our scientific liberty and freedom of choice when it comes to selecting our writing method and results presentation style. As long as we adhere to the journal's instructions and guidelines, we will continue to exercise this freedom in our research work. |
1. “The abstract need to refer to the obtained qualitative results as well quantitative results. Giving the fitting goodness without a necessary background is not informative. Therefore, it is recommended to correct abstract’’. |
The authors appreciate the comment. The abstract has been revised to show quantitative results. (Page 2, line 12) “At a 5% concentration, the main hydration peak for GU cement emerged after 91 hours, whereas for CSA cement, it appeared much earlier, at 2.5 hours.”
(Page 2, line 20) “Following 28 days of hydration, the compressive strength values for CSA cement were 36.7%, 63.5%, and 71% higher than those of GU cement at hemp hurd lixiviate concentrations of 0.5%, 2%, and 5%, respectively.” |
2- Introduction. In the first section there is no in-depth discussion of the behavior of cementitious composites modified by supplementary cementitious materials (SCMs) like fly ash or silica fume. Moreover, synergy between these materials is also very important in regard to reduce carbon footprint. There are recent articles published on this subject that should be discussed and referenced, such as: - „Compound Effects of Sodium Chloride and Gypsum on the Compressive Strength and Sulfate Resistance of Slag-Based Geopolymer Concrete”, Buildings 2023.
- “Combined effect of coal fly ash (CFA) and nanosilica (nS) on the strength parameters and microstructural properties of eco-friendly concrete”, Energies 2023.
- „Synergetic Effect of Superabsorbent Polymer and CaO-Based Expansive Agent on Mitigating Autogenous Shrinkage of UHPC Matrix, Materials 2023. |
The authors appreciate the comment. Then, the articles was discussed and referenced on the introduction on page 3 line 8.
“Additionally, cementitious composites modified with supplementary cementitious materials like fly ash or silica fume have the potential to reduce CO2 emissions in the atmosphere. Wei et al. [4] conducted a study on the influence of sodium chloride and gypsum on the compressive strength and sulfate resistance of slag-based geopolymer concrete. They found that by replacing 32.5% of slag Portland cement with sodium chloride, gypsum, and slag at proportions of 4:7, 5:13, and 5:75 in plain concrete, the cost and carbon emissions of geopolymer concrete were reduced by 25% and 48%, respectively. In another study, Grzegorz Ludwik [5] evaluated the effects of coal fly ash and nanosilica on the strength parameters and microstructural properties of eco-friendly concrete. The research demonstrated that using tailored blended cements composed of nanosilica and coal fly ash, with content up to 30% replacement level, significantly improved the parameters of the concrete composite. Moreover, this approach helped reduce the carbon footprint of cement-based materials, representing a positive step towards the production of eco-friendly concretes.”
|
3- Experiments. Please provide photos showing specimens during preparations and testing. Moreover, please provide more details regarding equipment used in the studies |
The authors express their gratitude for the comment received. In the methodology section, they have included sufficient information to enable the reproduction of their work. While providing photos can certainly enhance the readers' understanding of the techniques, it is not a common practice to offer a step-by-step guide. The authors are confident that they have furnished essential details for those interested in validating their work. For those curious about specific procedures, the appendix on page 41 showcases photos of specimens and the equipment used in this study. 1. Hot water extraction. 2. Filtration 3. evaporation of water
4. Hemp powder for the preparation of hemp hurds lixiviate at a concentration of 0.5%, 2% and 5%.
5. Cement paste preparation for compression tests
6. RIELLE compression/tension tester
7. TA instrument for TG analysis.
8. X-ray diffraction measurement.
9. Scanning Electron Microscopy
|
4- Results. All charts with test results should be presented in colors. |
The authors appreciate this comment. To ensure consistency in our result presentations, all graphs are displayed in color format. |
5- SEM studies. SEM images from figs. 9 and 10 are too small. In addition, all photos should have scale and magnification clearly marked. |
The scales and magnification in all SEM images on page 31 and 32, for Figure 9, 10 and Figure 11, were clearly indicated. |
6- Conclusions. Conclusions should be changed. Conclusions should be presented in several strong bullets. |
The authors believe that this concern has been previously addressed. They made a decision to adhere to their own writing style in this study. |

Reviewer 4 Report
The paper’s topic is an analysis of the influence of hemp hurd lixiviates on the cement composites properties. The problem is interesting, up-to-date, and connected to the need to develop low-emission building materials.
Since the bio-material releases monosaccharides, the observed delaying of cement hydration and worsening compressive strength follow the expectations and literature data. Therefore, the results are not very original. Also, using CSA cement instead of ordinary Portland cement is familiar. The strong point of the proposed paper is the wide range of the conducted tests (XRD, calorimetry, SEM, mechanical testing), which enabled the development and discussion of the mechanisms that convincingly explain the details of the observed phenomena.
The paper is comprehensive and well-prepared. I recommend publishing it; however, some corrections and completions are needed, namely:
- the last sentence of the Introduction says that “the investigation aims to determine the optimal parameters for pre-treatment processes that can enhance the compatibility between bio-aggregates and cementitious matrices”. It seems that this promise is not addressed in the paper and should be removed;
- the physical properties of the GU cement are missing, like the class of cement, density, etc.;
- the source of hemp hurd chemical composition should be given, the producer’s data or the Authors’ analysis,
- there is a mistake in the aluminate ion formula on pages 12 and 17; it cannot be AlO2- (one oxygen atom, two negative charges), but AlO2- (two oxygen atoms, one negative charge).
With the above amendments, the paper can be published in Materials.
Author Response
Comment |
Response/Amended Manuscript (Highlighted in Light Blue) |
The paper’s topic an analysis of the influence of hemp hurd lixiviates on the cement composites properties. The problem is interesting, up-to-date, and connected to the need to develop low-emission building materials. Since the bio-material releases monosaccharides, the observed delaying of cement hydration and worsening compressive strength follow the expectations and literature data. Therefore, the results are not very original. Also, using CSA cement instead of ordinary Portland cement is familiar. The strong point of the proposed paper is the wide range of the conducted tests (XRD, calorimetry, SEM, mechanical testing), which enabled the development and discussion of the mechanisms that convincingly explain the details of the observed phenomena. The paper is comprehensive and well-prepared. I recommended publishing it, however, some corrections and completions are needed, namely: |
The authors appreciate the comment and feedback.
|
1- the last sentence of the Introduction says that “the investigation aims to determine the optimal parameters for pre-treatment processes that can enhance the compatibility between bio-aggregates and cementitious matrices”. It seems that this promise is not addressed in the paper and should be removed. |
The authors appreciate this comment. The statement was removed.
|
2- the physical properties of the GU cement are missing, like the class of cement, density, etc. |
The statement has been amended on page 8, line 1 and revised as:
“The GU cement type is CEM I 42.5R, with a density of 3.15 and specific surface area of 393 m2/kg” |
3- the source of hemp hurd chemical composition should be given, the producer’s data or the Authors’ analysis. |
The statement has been amended on page 10, Table 2 and revised as: “Authors’analysis %’ |
4- there is a mistake in the aluminate ion formula on pages 12 and 17; it cannot be AlO2- (one oxygen atom, two negative charges), but AlO2- (two oxygen atoms, one negative charge). |
The authors thank the reviewer for this comment.
The formula on pages 23 and 30 was corrected as “AlO2- “
|

Round 2
Reviewer 3 Report
The paper is well revised. I have no further comments.